



# Pliocene evolution of the tropical Atlantic thermocline depth

Carolien M. H. van der Weijst[1], Josse Winkelhorst[1], Wesley de Nooijer[1], Anna von der Heydt[2], Gert-Jan Reichart[1,3], Francesca Sangiorgi[1], Appy Sluijs[1]

[1]Department of Earth Sciences, Utrecht University, 3584 CB Utrecht, the Netherlands
[2]Department of Physics, Utrecht University, 3584CC Utrecht, the Netherlands
[3]NIOZ Royal Netherlands Institute for Sea Research, 1797 SZ 't Horntje, the Netherlands

*Correspondence to*: C.M.H. van der Weijst (c.m.h.vanderweijst@uu.nl)

**Abstract.**

It has been hypothesized that global temperature trends are tightly linked to tropical thermocline depth, and that thermocline shoaling played a crucial role in the intensification of late Pliocene northern hemisphere glaciation. The Pliocene thermocline evolution in the Pacific Ocean is well documented and supports this hypothesis, but thermocline records from the tropical Atlantic Ocean are limited. We present new planktonic foraminiferal Mg/Ca, $\delta^{18}O$ and $\delta^{13}C$ records from the late Pliocene

interval at Ocean Drilling Program Site 959 in the eastern equatorial Atlantic (EEA), which we use to reconstruct ocean temperatures and relative changes in salinity and thermocline depth. Data were generated using surface-dwelling *Globigerinoides ruber* and subsurface-dwelling *Neogloboquadrina dutertrei*. Reduced gradients between the surface and subsurface records indicate deepening of the EEA thermocline at the end of the Mid-Piacenzian Warm Period (mPWP; ~3.3-3.0 Ma). We connect our late Pliocene records to previously published early Pliocene $\delta^{18}O$ data from Site 959 and compare

these to the Site 1000 in the Caribbean Sea. Over the course of the Pliocene, thermocline changes in the EEA and Caribbean Sea follow similar patterns, with prominent step-wise thermocline deepening between ~5.5 and 4.0 Ma, gradual shoaling up to the mPWP, followed by minor deepening at the end of the mPWP. The tropical thermocline depth evolution of the tropical Atlantic differs from the Pacific, which is characterized by gradual basin-wide shoaling across the Pliocene. These results potentially challenge the hypothesized link between tropical thermocline depth and global climate. The mechanisms behind

the periodically divergent Pacific and Atlantic thermocline movements remain speculative. We suggest that they are related to basin geometry and heterogenous temperature evolutions in regions from where thermocline waters are sourced. A positive feedback loop between source region temperature and tropical cyclone activity may have amplified tropical thermocline adjustments.



## 1 Introduction

The Pliocene (5.33-2.58 Ma) is the most recent geological epoch with substantially higher greenhouse gas concentrations (de la Vega et al., 2020; Martínez-Botí et al., 2015; Stap et al., 2016) and elevated global surface temperatures (Brierley et al., 2009; Dowsett et al., 2016; McClymont et al., 2020) compared to preindustrial times. This makes it an interesting interval to study for its potential analogies with our future climate (Burke et al., 2018). Typical features of the warm Pliocene ocean are the poleward expansion of the tropical warm pools and a reduction of the zonal sea surface temperature (SST) gradient in the Pacific (Brierley et al., 2009; Dekens et al., 2008; Fedorov et al., 2013). Another key feature of the Pliocene ocean is the deep tropical Pacific thermocline that gradually shoaled towards the end of the Pliocene (Dekens et al., 2007; Ford et al., 2012, 2015; Steph et al., 2010), while the meridional and zonal SST gradient steepened (Fedorov et al., 2015). It has been suggested that this shoaling reached a critical threshold around 3 Ma and played an important role in the onset of northern hemisphere glaciation (Fedorov et al., 2006; Philander and Fedorov, 2003; Steph et al., 2010). Moreover, because there is general coherence between Pliocene changes in meridional, zonal, and vertical (i.e. thermocline depth) temperature gradients in the Pacific, Fedorov et al. (2015) suggested that these temperature gradients are somehow mechanistically linked.

Interestingly, while the thermocline shoaled across the tropical Pacific, $\delta^{18}O$ and Mg/Ca records of planktic foraminifera show that the Caribbean thermocline deepened during the early Pliocene (Steph et al., 2010). This was interpreted by Steph et al. (2010) as a local phenomenon. However, early Pliocene $\delta^{18}O$ records form the Eastern Equatorial Atlantic (EEA) also show a thermocline deepening (Norris, 1998). This suggests that the entire tropical Atlantic may have had a different early Pliocene thermocline evolution than the Pacific. If so, the link between tropical thermocline depth and Pliocene global temperature trends may need to be reconsidered. Late Pliocene thermocline records from the EEA are currently unavailable, yet necessary for a complete evaluation of Pliocene tropical thermocline depth evolution in a context of global climate changes.

We present the first records of late Pliocene thermocline depth variability in the EEA. We generated $\delta^{18}O$, $\delta^{13}C$ and Mg/Ca records on surface-dwelling *Globigerinoides ruber* and subsurface-dwelling *Neogloboquadrina dutertrei* from Ocean Drilling Program (ODP) Site 959 sediments. We use the offsets between these records to infer changes in thermocline/nutricline depth and relative salinity changes. We evaluate our new records against previously published thermocline depth records from the Atlantic and Pacific to evaluate global patterns of thermocline depth changes across the Pliocene.

## 2 Methods

### 2.1 Site and modern oceanographic setting

We used sediments recovered at ODP Site 959 during Leg 159 in the Gulf of Guinea at ~160 km offshore Ghana and Ivory Coast (3.62°N, 2.73°W; 2090 m depth; Mascle et al., 1996, Figure 1). At present, this region is characterized by a shallow



thermocline compared to the west Atlantic (Figure 2). Surface ocean circulation at Site 959 is characterized by the eastward flowing geostrophic Guinea Current that originates from the North Equatorial Counter Current and the Canary Current (Figure 1). The thermal structure of the local water column varies seasonally in response to monsoon-induced changes in upwelling intensity (Figure 1b). In West Africa, the intertropical convergence zone (ITCZ) shifts from 5-10°N in February to 15-20°N

in August, which forces strong southwesterly monsoonal winds over the Gulf of Guinea in boreal summer (Gu and Adler, 2004). A major upwelling event occurs along the northern coast in summer, which is likely forced by a combination of local wind-stress, wind-induced eastward propagating Kelvin waves and intensification of the Guinea Current (Djakouré et al., 2017; Verstraete, 1992), which together raise the thermocline. A shorter and weaker coastal upwelling event typically occurs in winter (Wiafe and Nyadjro, 2015). South of Site 959, upwelling occurs at the equator in response to equatorial divergence and

along the African west coast due to persistent Ekman-pumping (Bakun and Nelson, 1991; Figure 1).

## 2.2 Thermocline depth reconstructions

The thermocline is typically defined as the depth at which the vertical temperature change is at its steepest. This depth can be approximated with the 20°C isotherm in the modern ocean, which is typically done in modelling studies, but different isotherms

may be more appropriate in warmer climates (e.g. Von der Heydt and Dijkstra, 2011). A common approach for qualitative reconstructions of past thermocline depth variability is by paired tracing of surface ocean and shallow subsurface ocean conditions. We trace surface ocean conditions with *Globigerinoides ruber* and subsurface conditions with *Neogloboquadrina dutertrei,* which typically reside in the upper ocean and at thermocline depths respectively (Anand et al., 2003; Rebotim et al., 2017; Steph et al., 2009). Figure 3 conceptually shows how thermocline depth determines tracer offsets between surface- and

subsurface-dwellers (Δsurface-subsurface). We use foraminiferal Δsurface-subsurface $\delta^{18}$O, Mg/Ca, and $\delta^{13}$C (hereafter $\Delta\delta^{18}$O, $\Delta$Mg/Ca and $\Delta\delta^{13}$C) to trace changes in thermocline and nutricline depth, and paired $\Delta\delta^{18}$O-$\Delta$Mg/Ca to trace relative salinity changes.

## 2.3 Stable isotope analysis

Samples were taken at 5-20 cm intervals from cores 959C-5H and 959C-6H between 35.77 and 48.73 revised Meters Composite Depth (rMCD; splice by Vallé et al., 2016). The studied interval of nannofossil/foraminifer ooze (Mascle et al., 1996) was dated using oxygen isotope stratigraphy (van der Weijst et al., 2020) and spans 3.5-2.8 Ma. The preservation of planktonic foraminifera in the Pliocene of 959 is generally very good; a large proportion of the tests have a "glassy" appearance, and *G. ruber* often still has spines around its aperture (Figure S1). These features suggest that diagenetic calcite overgrowth,

dissolution and recrystallisation are minimal (Edgar et al., 2015).





For each measurement, 50-60 foraminiferal tests (*G. ruber* white s.s. and *N. dutertrei)* were picked from the 250-355 µm size fraction. Cleaning of *G. ruber* was performed at Utrecht University and that of *N. dutertrei* at the Royal Netherlands Institute for Sea Research (NIOZ) following slightly different protocols to remove clay and coccoliths. After gently crushing the tests

between two glass plates, *G. ruber* was ultrasonically cleaned with ethanol and rinsed with distilled water. Crushed tests of *N. dutertrei* were ultrasonically cleaned three times with methanol, two times with hot (95°C) 1% NaOH/$H_2O_2$ solution and rinsed with Milli-Q in between each step. Stable isotope analyses were performed at Utrecht University on a Thermo Finnigan GasBench-II carbonate preparation device coupled to a Thermo Finnigan Delta-V mass spectrometer. The international IAEA-CO-1 standard was measured to calibrate to the Vienna Pee Dee Belemnite (VPDB) and the inhouse NAXOS standard to

correct for drift and track analytical precision, which was better than 0.1‰ for both carbon and oxygen isotopes.

**2.4 Mg/Ca analysis and calibration**

Sediment samples were wet-sieved with tap water and dried overnight at a low temperature. Per sample, between 20 and 60 specimens of the surface-dwelling *G. ruber* (white, s.s.) and the thermocline-dwelling *N. dutertrei* were picked form the 250-

355 µm size fraction and cleaned following the protocol of Barker et al. (2003). Tests were gently crushed between two glass plates and ultrasonically treated with methanol (3x), hot 1% NaOH/$H_2O_2$ solution at 95°C (2x), 0.001M $HNO_3$ (1x) and rinsed with Milli-Q in between each step. Trace element analyses were performed at the Royal Netherlands Institute for Sea Research (NIOZ) on an Element-2 ICP-MS. Note that the four M2 samples (identified by peak benthic $\delta^{18}O$; Van der Weijst et al., 2020) were also used in a separate analysis for which a large quantity of carbonate was needed. For this reason, specimens of *G.*

*ruber* from samples 959C/6H-2/102-104, 959C/6H-2/107-109, 959C/6H-2/112-114 and 959C/6H-2/117-119 were combined to a "M2 bulk sample", and only 959C/6H-2/102-104 could also be analyzed separately. We calculate ocean temperatures using Mg/Ca values (mmol/mol).

Ambient seawater temperature is typically the dominant control on the ratio of magnesium to calcium in foraminiferal shells

(e.g. Hönisch et al., 2013; Tierney and Malevich, 2019). At present, there is no standard approach to calibrate Mg/Ca to temperature, which complicates site-to-site comparisons (McClymont et al., 2020). However, there is general consensus that calibrating Mg/Ca from fossil foraminifera to paleotemperature requires corrections for preferential removal of Mg over Ca during dissolution of calcite in the water column and sediment (Dekens et al., 2002; Regenberg et al., 2006) and the effect of the changing Mg/Ca of seawater (Mg/Ca$_{sw}$) on geological timescales in relation to rates of oceanic crust production (Hardie,

1996). Moreover, most calibrations assume a constant partitioning coefficient ($D_{Mg}$) at a given temperature, i.e., a linear relationship between Mg/Ca$_{sw}$ and calcite Mg/Ca (Mg/Ca$_{calcite}$), but experiments suggest that $D_{Mg}$ changes in response to Mg/Ca$_{sw}$ (De Nooijer et al., 2017; Evans et al., 2016; Evans and Müller, 2012; Ries, 2004). To improve calibrations, species-specific $D_{Mg}$ sensitivities and past Mg/Ca$_{sw}$ need better constraints.



Here, we apply a constant Pliocene Mg/Ca$_{sw}$ correction to the species-specific core top calibrations from Dekens et al. (2002), in which a constant partitioning coefficient is assumed, and a dissolution correction is embedded based on species, ocean basin and core depth:

$$Mg/Ca \text{ (calcite)} = B \cdot \frac{Past\ Mg/Ca_{sw}}{Present\ Mg/Ca_{sw}} \cdot e^{A(T - C \cdot core\ depth)} \qquad (1).$$


The slope (A) is 0.09 for *G. ruber* and 0.08 for *N. dutertrei*, the intersect (B) is 0.38 for *G. ruber* and 0.60 for *N. dutertrei,* and the dissolution parameter (C) is 0.61 for *G. ruber* and 2.8 for *N. dutertrei* (Dekens et al., 2002). The present-day Mg/Ca$_{sw}$ is 5.2 mmol/mol and we use the Pliocene estimate of 4.3 mmol/mol (Evans et al., 2016). The core depth at Site 959 is 2.09 km. Note that *G. ruber* is calibrated to SST and *N. dutertrei* to 50 m depth subsurface temperature respectively (Dekens et al.,

135 2002).

**3 Results**

Late Pliocene Mg/Ca SST estimates from surface dwelling *G. ruber* vary between 26.5°C and 29.0°C and Mg/Ca subsurface temperatures from *N. dutertrei* between 21.0°C and 25.4°C (Figure 4). Surface $\delta^{18}$O varies largely between -2.18‰ and -

1.31‰ and subsurface $\delta^{18}$O between -1.19‰ and -0.14‰. Surface $\delta^{18}$O and Mg/Ca barely show a long-term trend, although both record distinct cooling during the M2 glacial and peak temperatures during the KM3 interglacial. Subsurface $\delta^{18}$O and Mg/Ca both show a warming trend following the M2 glacial, although it is more gradual in Mg/Ca-SubST than in $\delta^{18}$O-SubST. Surface and subsurface $\delta^{13}$C values vary between 0.91-2.13‰ and 0.56-1.45‰ respectively. Both show an increasing trend in the older part of the interval, and abruptly drop around KM3. In the younger part of the interval, surface $\delta^{13}$C remains stable,

while subsurface $\delta^{13}$C increases.

The $\Delta$(surface-subsurface) Mg/Ca, $\delta^{18}$O and $\delta^{13}$C records (Figure 4) follow a similar trend towards smaller offsets. Both $\Delta$Mg/Ca and $\Delta\delta^{18}$O signal a vertical temperature gradient decrease from ~5.0 to 3.5°C. The $\Delta\delta^{18}$O record decreases after the M2 glacial, thereby leading the $\Delta$Mg/Ca and $\Delta\delta^{13}$C decrease around the KM3 interglacial by ~150 kyr. Moreover, $\Delta\delta^{18}$O

stabilizes in the younger part of the interval, while $\Delta$Mg/Ca and $\Delta\delta^{13}$C continue to decrease.



## 4 Discussion

### 4.1 Late Pliocene thermocline, nutricline and salinity changes in the Eastern Equatorial Atlantic

The decreasing surface-subsurface Mg/Ca and $\delta^{18}$O gradients between 3.3 Ma and 2.8 Ma at Site 959 reflect thermocline deepening in the eastern equatorial Atlantic (Figure 4). These decreasing vertical gradients are driven by a gradual warming
trend in the subsurface ocean: *G. ruber* surface ocean Mg/Ca and $\delta^{18}$O records remain stable across the interval, whereas the *N. dutertrei* subsurface records show an increase in Mg/Ca and decrease in $\delta^{18}$O following the M2 glacial. Crude $\delta^{18}$O-based temperature estimates calculated with the linear Shackleton (1974) calibration have similar values and amplitude of variation as the Mg/Ca estimates, but the decreasing $\Delta\delta^{18}$O trend leads decreasing $\Delta$Mg/Ca (Figure 4).

Besides analytical and calibration errors, several confounding factors limit the use of $\Delta\delta^{18}$O and $\Delta$Mg/Ca as quantitative proxies for thermocline depth, including: vertical migration of foraminifera in the water column (e.g. Steph et al., 2009), variability in the mass and/or preservation of the high-$\delta^{18}$O and low-Mg/Ca secondary calcite crust that *N. dutertrei* precipitates at the end of its lifecycle (Jonkers et al., 2012; Steinhardt et al., 2015), seawater carbon chemistry (Spero et al., 1997) and light intensity (Spero and Michael, 1987). Moreover, a discrepancy is observed in depth habitat ranges of planktonic species based
on tows versus those reconstructed based on their $\delta^{18}$O values in core tops (Rebotim et al., 2017). Crucially, however, $\Delta\delta^{18}$O and $\Delta$Mg/Ca perform well in predicting larger thermocline patterns in the tropical Atlantic (Anand et al., 2003; Steph et al., 2009).

In addition to ambient temperature, calcite $\delta^{18}$O is determined by local seawater $\delta^{18}$O ($\delta$w), which is in turn tightly linked to
salinity (Bigg and Rohling, 2000). Recently, Dämmer et al. (2020) advised against using coupled Mg/Ca-$\delta^{18}$O data for salinity reconstructions, because cumulative uncertainties from aforementioned confounding factors are too large to reliably reconstruct measured $\delta$w. However, these uncertainties are mostly relevant when applying this method quantitatively to discrete samples. To reduce these uncertainties, we use a semi-quantitative approach to assess regional long-term salinity trends by calculating the relative vertical salinity gradient ($\Delta$salinity) from the discrepancy between $\Delta$Mg/Ca and $\Delta\delta^{18}$O (Figure
5). We assume that $\Delta$Mg/Ca best approaches $\Delta$temperature, despite the salinity effect on Mg/Ca (Gray et al., 2018; Hönisch et al., 2013), and that the residual $\Delta\delta^{18}$O is explained by salinity changes. Note that the intersects of the $\Delta$Mg/Ca and $\Delta\delta^{18}$O temperature reconstructions in Figure 5 may not correspond to a vertical salinity gradient of 0 due to uncertainties in both the $\delta^{18}$O-paleotemperature and the Mg/Ca calibrations. Rather, the intersects indicate an unknown baseline $\Delta$salinity to which the rest of the record can be compared.


Absolute estimates of salinity from residual $\Delta\delta^{18}$O values (Figure 5) depend on the assumed slope of the linear $\delta$w-salinity ($\delta$w-S) relationship. In the modern tropical Pacific, the $\delta$w-S slope varies with a factor >3 (0.09-0.32‰/salinity unit) between across the equatorial Pacific, because the precipitation-forced isotopic "amount effect" (lower $\delta$w with higher tropical



precipitation rates), is stronger in the western than eastern Pacific (Conroy et al., 2017). In the tropical Atlantic, however, this
relationship is poorly documented. Using the slope provided by Craig and Gordon (1965; 0.11‰/salinity unit), we calculate
that Δsalinity was reduced by ~2 salinity units during the Mid Piacenzian Warm Period (mPWP; 3.264-3.025 Ma) compared
to the bordering intervals (Figure 5). At present, the vertical salinity gradient is ~1 in the Gulf of Guinea (Figure 1b), suggesting
that the Pliocene baseline salinity gradient was stronger than at present and/or that the $\delta$w-S slope was steeper. A doubling of
the assumed tropical Atlantic $\delta$w-S slope (0.22‰/salinity unit) is needed to solve the Δsalinity peak to the modern gradient of
~1. It falls within range of modern tropical Pacific slopes and may be realistic for the interglacial Pliocene Atlantic, which was
characterized by intensified monsoonal precipitation over West Africa (Haywood et al., 2020).

In the absence of major circulation changes, the $\Delta\delta^{13}$C record mainly reflects changes in the biological carbon pump and
nutricline depth, which in the modern tropical ocean is tightly coupled to thermocline depth (Wilson and Adamec, 2002). We
here consider coupled thermocline and nutricline deepening to be the most likely explanation for the observed synchronous
changes in ΔMg/Ca and $\Delta\delta^{13}$C around KM3. The relative lead of the $\Delta\delta^{18}$O decrease may explained by subsurface freshening
(i.e., deepening of the halocline, Figure 2b) in response to increased precipitation rates during the mPWP. This implies that
ΔMg/Ca may outperform $\Delta\delta^{18}$O as a proxy for thermocline depth, but salinity changes would have to be strong in order to
fully obscure the thermocline depth signal in the $\Delta\delta^{18}$O record. Despite temporarily increased precipitation during the mPWP,
the net Δtemperature estimates from $\Delta\delta^{18}$O and ΔMg/Ca are equal between 3.5 and 2.8 Ma.

**4.2 Divergent evolutions of the tropical Atlantic and Pacific thermocline**

Our results show that the eastern equatorial Atlantic thermocline and nutricline deepened following the warmest mPWP
interglacials, while SST and deep ocean records show global cooling (Herbert et al., 2016; Lisiecki and Raymo, 2005). Late
Pliocene thermocline and nutricline changes are similar at Site 959 in the eastern equatorial Atlantic and Site 1000 in the
western tropical Atlantic (Figure 5), indicating that tropical thermocline deepening occurred across the entire basin. To further
explore global patterns of Pliocene tropical thermocline movements, we connect our new Site 959 $\delta^{18}$O data to the early
Pliocene records of Norris (1998) in Figure 6. Across the Pliocene, thermocline depth evolution was similar in the eastern
equatorial Atlantic and Caribbean Sea (Figure 6). The late Pliocene ~0.4‰ $\Delta\delta^{18}$O decrease at Site 959 and Site 1000 is
relatively small compared to the fluctuations in the latest Miocene and early Pliocene (Figure 6). Both sites show a net ~1.5‰
$\Delta\delta^{18}$O decrease between 5.5 and 4.0 Ma, indicating significant thermocline deepening. This is followed by an interval of
moderate thermocline shoaling, before deepening again in the late Pliocene. With the exception of the late Pliocene interval at
Site 1241 (Figure 6), tropical Pacific proxy records reflect overall Pliocene thermocline shoaling (Ford et al., 2012, 2015;
LaRiviere et al., 2012; Steph et al., 2006b, 2010). Although changes in $\delta^{18}$O may not always be proportional to vertical
movement of the thermocline (see Section 4.1), the early Pliocene fluctuations in the Atlantic records are appreciably larger





than the long-term shifts towards lower $\Delta\delta^{18}O$ values in the Pacific (Figure 6). This suggests that the tropical Atlantic thermocline underwent major changes during the early Pliocene, which were not paralleled in magnitude by late Pliocene changes in the Atlantic or by any Pliocene thermocline change in the Pacific.


It was suggested by Fedorov et al. (2015) that global temperature trends are tightly linked to tropical thermocline depth and that thermocline shoaling played a crucial role in the onset of Northern Hemisphere glaciations around 3 Ma Fedorov et al., 2006). Heat that is gained in the tropics must be balanced by heat loss at high latitudes, and because more heat can be gained by an ocean with a shallow thermocline, tropical thermocline shoaling should lead to surface ocean cooling at high latitudes
(Boccaletti et al., 2004; Fedorov et al., 2006; Philander and Fedorov, 2003). While this theory is supported by the majority of the Pacific thermocline records (Ford et al., 2012, 2015; LaRiviere et al., 2012; Steph et al., 2006b, 2010), it appears to be inconsistent with basin-wide deepening of the tropical Atlantic thermocline (Figure 6). The link between ocean SST gradients, tropical thermocline depth, and ocean heat transport may be different in the Atlantic and Pacific oceans, because the asymmetric geometry of the Atlantic basin leads to a different pattern of tropical thermocline ventilation (Harper, 2000). Also,
Atlantic Meridional Overturning Circulation (AMOC) leads to northward ocean heat transport in both hemispheres, in contrast to the Pacific Ocean (Forget and Ferreira, 2019). However, because the tropical Pacific is larger than the tropical Atlantic, Pacific thermocline shoaling may have been sufficient to balance heat loss at high latitudes, even if partly counteracted by Atlantic thermocline deepening.

**4.3 Controls on Pliocene thermocline changes**

**4.3.1 Central American Seaway closure**

It has been hypothesized that CAS closure played a major role in the global tropical thermocline depth (Steph et al., 2010; Zhang et al., 2012). In this scenario, salt transport to the North Atlantic increased as a consequence of reduced inflow of relatively fresh Pacific surface waters, which in model simulations promotes the production of NADW (Lunt et al., 2007;
Sepulchre et al., 2014; Steph et al., 2010; Zhang et al., 2012). Steph et al. (2010) reasoned that an increased volume of NADW was associated with a greater volume of the "cold water sphere", which raised the tropical thermocline everywhere except for the Caribbean region. There, reduced inflow of cold Pacific waters caused the thermocline to deepen locally. However, our study shows that tropical thermocline deepening occurred across the basin (Figure 6). Even if CAS closure had promoted NADW production and/or AMOC strength, it is not clear how this, in turn, would have affected tropical thermocline depth, as
both negative (Lopes dos Santos et al., 2010) and positive (Venancio et al., 2018) relationships between thermocline depth and AMOC strength have been inferred from proxy data. Moreover, it is unclear if CAS closure could have had an opposite effect on tropical Atlantic and Pacific thermocline depths, as this conflicts with modeling studies (Steph et al., 2010).



### 4.3.2 Source region SSTs and tropical cyclones

Tropical thermocline waters are sourced from mid-latitude surface waters (Harper, 2000). It has therefore been suggested that extratropical cooling contributed to the Pliocene shoaling of the tropical Pacific thermocline (Ford et al., 2012, 2015). During the early Pliocene, midlatitude SSTs were indeed generally higher than preindustrial (Brierley et al., 2009). However, Pliocene midlatitude temperature evolutions were regionally variable (Herbert et al., 2016), with a notable asymmetry between the northern and southern hemisphere (Pontes et al., 2020). At present, Pacific thermocline waters are sourced from mid-latitude

surface waters in both the northern and southern hemisphere, whereas in the Atlantic, the thermocline is predominantly sourced from the southern hemisphere as a consequence of the asymmetric basin geometry (Harper, 2000). Mid-latitude SST records from potential Pacific tropical thermocline source regions register predominantly cooling between 5.5 and 2.8 Ma. In contrast, South Atlantic Site 1088 registers warming during the latest Miocene/early Pliocene, and during the mPWP (Figure 7), in tandem with tropical Atlantic thermocline deepening. However, South Atlantic cooling between ~4.5 and 3.5 Ma (Figure 7) is

not mirrored by thermocline shoaling, suggesting that source region SST was not likely the only driver of tropical Atlantic thermocline depth.

Tropical thermocline depth is also affected by cyclone activity, which, in turn, is linked to the latitudinal SST gradient in a positive feedback mechanism (Fedorov et al., 2010). In other words, source region SSTs are potentially relevant to tropical

thermocline dynamics in terms of both stand-alone trends, and in the context of distant SST trends. Tropical cyclones force vertical mixing in the upper ocean, which deepens the tropical thermocline (Bueti et al., 2014; Jansen et al., 2010). Tropical cyclone activity was higher in the early Pliocene than at present, especially in the Pacific (Fedorov et al., 2010). Gradually increasing zonal and meridional SST gradients (Fedorov et al., 2015) would have promoted stronger Walker and Hadley circulation, thereby reducing tropical cyclone activity and raising the tropical thermocline (Brierley et al., 2009; Fedorov et

al., 2010). The Pacific underwent a stronger reduction in tropical cyclone activity than the Atlantic (Fedorov et al., 2010), which may be linked to the contrasting evolution of the Pacific and Atlantic tropical thermocline depth.

### 5 Conclusions

Our new Mg/Ca, $\delta^{18}$O and $\delta^{13}$C records from Site 959 indicate late Pliocene thermocline and nutricline deepening in the eastern

equatorial Atlantic, starting at the end of the mPWP. A temporal discrepancy between $\Delta\delta^{18}$O and $\Delta$Mg/Ca can be explained by a transient reduction of the vertical salinity gradient in response to intensification of west African monsoonal precipitation during the mPWP. Tropical thermocline deepening occurred across the Atlantic basin, as is indicated by a nearly identical $\Delta\delta^{18}$O record at Site 1000 in the Caribbean Sea. This seems inconsistent with the hypothesis that Pliocene global cooling was linked to tropical thermocline shoaling, as observed in the Pacific, although Atlantic thermocline adjustments may have had

less effect on the global heat budget because the Atlantic basin is smaller. The mechanisms driving Pliocene thermocline

changes are currently speculative. In climate models, Central American Seaway closure leads to global thermocline shoaling, but during most of the early and late Pliocene, opposite changes occurred in the tropical Atlantic and Pacific thermoclines. We therefore suggest to further explore if the opposite alternative mechanisms, such as source water temperature forcing in the Southern Ocean in combination with tropical cyclone activity. Due to basin morphology, tropical Atlantic thermocline waters

are sourced from the southern hemisphere, whereas both hemispheres feed the tropical thermocline waters in the Pacific. Asymmetric northern and southern hemisphere temperature trends may therefore have contributed to the differences in Atlantic and Pacific thermocline depth changes. A positive feedback loop between meridional SST gradients and tropical cyclones may have amplified vertical thermocline movements.

**Data availability**

New Site 959 data are available as a supplement to this paper and will be uploaded to the PANGAEA online data repository upon publication.

**Competing interests**

The authors declare that they have no conflict of interest.

**Acknowledgements**

We thank the International Ocean Discovery Program and the predecessors for samples and data, Arnold van Dijk (UU), Cindy Remijnse-Schrader (UU), Wim Boer (NIOZ) and Geert-Jan Brummer (NIOZ) for technical support and advice. We are also grateful to editor Luc Beaufort (Cerege) and two anonymous reviewers for constructive criticism on an earlier draft of this work. This work was carried out under the program of the Netherlands Earth System Science Centre (NESSC), financially supported by the Ministry of Education, Culture and Science (OCW).

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




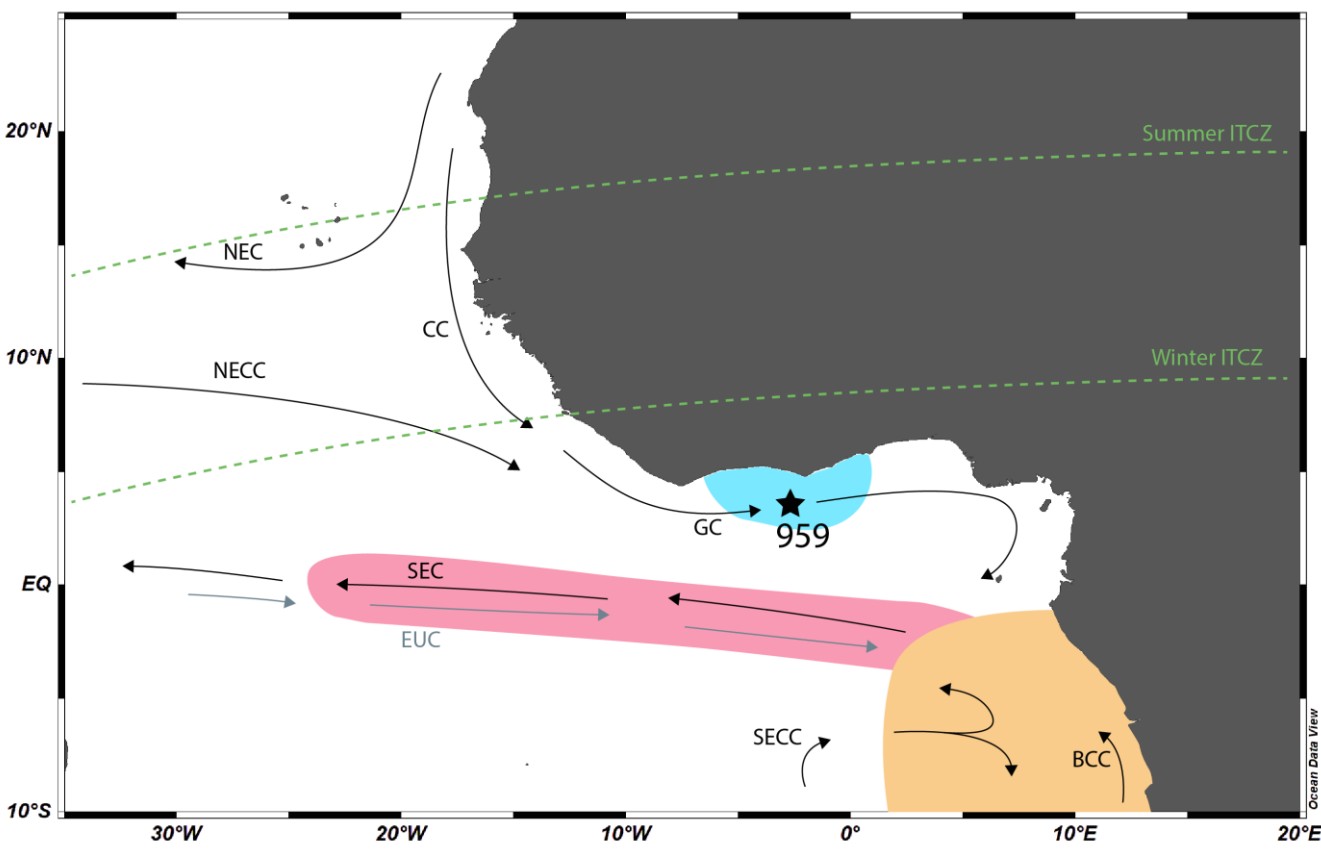

**Figure 1. General surface currents and upwelling areas in the eastern equatorial Atlantic. Blue shading: coastal upwelling in boreal summer; pink shading: equatorial upwelling; orange shading: permanent coastal upwelling. NEC: North Equatorial Current; NECC: North Equatorial Counter Current; CC: Canary Current; GC: Guinea Current; SEC: South Equatorial Current; EUC: Equatorial Undercurrent; SECC: South Equatorial Counter Current; BCC: Benguela Coastal Current. Green stippled lines mark the annual range of the Intertropical Convergence Zone (ITCZ). Figure after Norris, (1998); Wagner, (1998); Wiafe and Nyadjro (2015). Map generated with Ocean Data View (Schlitzer, 2020) (Schlitzer, Reiner, Ocean Data View, odv.awi.de, 2020).**





**Figure 2. (a): Upper ocean temperature (shaded contours; Locarnini et al., 2013) and salinity (black contours; Zweng et al., 2013) profiles across the black stippled line (see map) between Site 1000 and Site 959. The white stippled line shows the 20°C isotherm depth. Cross section generated with Ocean Data View (Schlitzer, 2020) (Schlitzer, Reiner, Ocean Data View, odv.awi.de, 2020). (b): Seasonal vertical temperature and salinity profiles in the Gulf of Guinea (2.5°N 3.5°W; Locarnini et al., 2013; Zweng et al., 2013). Symbols (left panel) mark the 20°C isotherm depth, which varies seasonally between ~40-60 meters.**



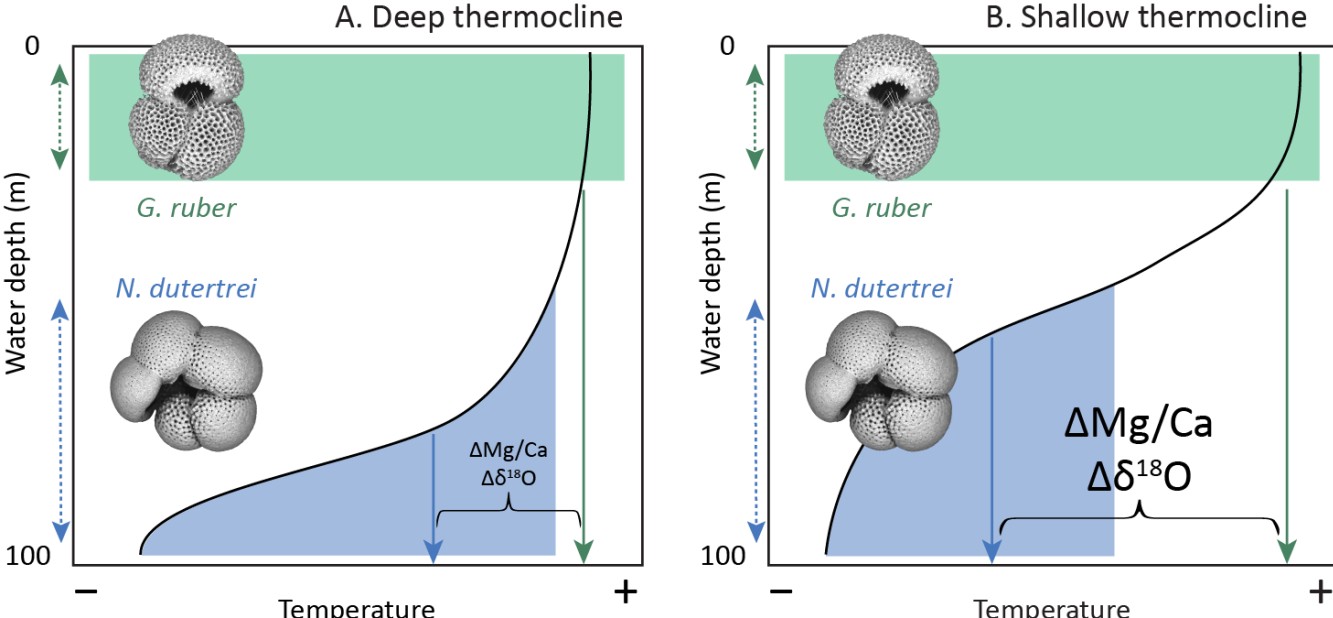

**Figure 3. Conceptual manifestation of the thermocline depth as a ΔMg/Ca Δδ$^{18}$O signal. Depth differentiation of the *G. ruber* and *N. dutertrei* habitats (stippled arrows) on a vertical temperature gradient (black line) sets a δ$^{18}$O and Mg/Ca offset in exported foraminiferal tests (solid arrows). The magnitude of this offset changes in response to thermocline shoaling and deepening.**



**Figure 4. (a):** Surface (*G. ruber*; blue) and subsurface (*N. dutertrei*; orange) $\delta^{18}O$ and Mg/Ca temperature **(b):** Surface (*G. ruber*; green) and subsurface (*N. dutertrei*; pink) $\delta^{13}C$ records. Bold lines in (a) and (b) are smoothed with a LOESS moving regression model. **(c):** $\Delta$(Surface-subsurface) offsets from (a) and (b). Site 959 benthic $\delta^{18}O$ record (van der Weijst et al., 2020) as grey silhouette for reference. Vertical bands indicate the M2 glacial and KM3 interglacial stage. Note that planktic $\delta^{18}O$ and temperature axes are evenly scaled according to the $\delta^{18}O$-T relationship of Shackleton (1974).





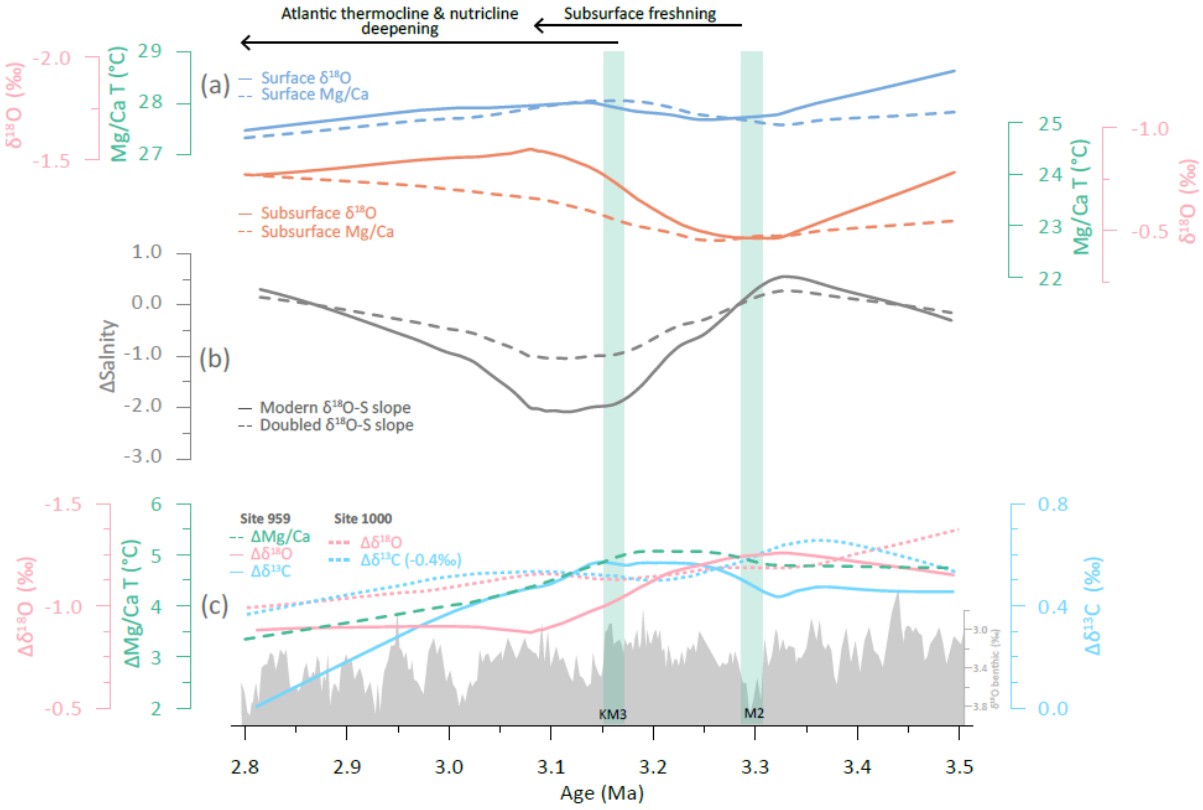

**Figure 5. (a): Smoothed Mg/Ca and δ¹⁸O records from Site 959 (b): Relative magnitude of the salinity gradient between surface and subsurface, calculated by deconvolving Δ(surface-subsurface) δ¹⁸O and Mg/Ca, and assuming a modern tropical Atlantic δ¹⁸O-S slope (0.11‰/salinity unit; solid line) and a doubled slope (dashed line). (c): LOESS smoothed thermocline and nutricline tracers from Site 959 (as in Figure 4c) and Site 1000 in the Caribbean Sea (stippled lines, data from Steph, 2005). Site 1000 Δδ¹³C was steeper than at Site 959, and was corrected with -0.4‰ to aid visual comparison.**

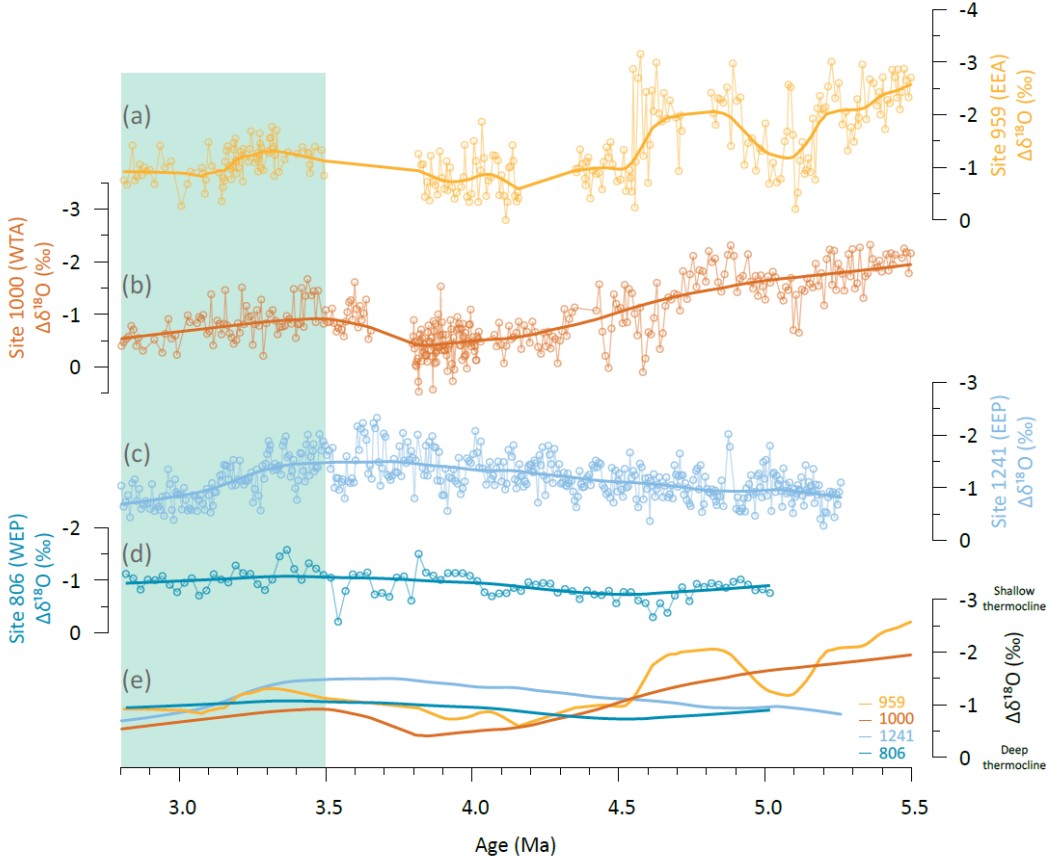

525

**Figure 6. Compilation of tropical Δδ¹⁸O data. (a): Eastern Equatorial Atlantic (EEA) Site 959, this study and Norris (1998). (b): Western Tropical Atlantic (WTA) Site 1000, Steph, (2005) and Steph et al. (2006a). (c): Eastern Equatorial Pacific (EEP) Site 1241, Steph et al. (2006b). (d): Western Equatorial Pacific (WEP) Site 806, LaRiviere et al. (2012). (e) LOESS smoothed records. Surface *Trilobatus sacculifer* records were corrected with +0.33‰ to *G. ruber* based on their offset in the modern ocean (Steph et al., 2009).**
530  **Blue square marks late Pliocene interval covered in Figures 4 and 5.**



**Figure 7. Regionally sorted mid-latitude $U^{k'}_{37}$ temperature records from the Herbert et al. (2016) compilation. (a): North Atlantic; (b): South Atlantic; (c): North Pacific; (d): South Atlantic. Vertical shading: intervals with consistent tropical Atlantic thermocline deepening (yellow) and shoaling (green).**

535