# Peer review of "Pliocene evolution of the tropical Atlantic thermocline depth"

_Climate of the Past, 2021_

## Author Comment (AC1)

Dear Editor,

We thank Anonymous Referee #1 for investing their time in evaluating our manuscript again. We are very happy with the positive evaluation and are very happy to discuss the remaining points here and in a revised manuscript.

*Review van der Weijst et al ' Pliocene evolution of the tropical Atlantic thermocline depth. This is a revised manuscript submitted to Clim. Past. Discuss in 2020 which I also reviewed. For this revised manuscript the authors carried out further planktonic foraminifera Mg/Ca measurements and also show the carbon isotopes (new Figures 3 and 4). Thermocline trends inferred from DMg/Ca, Dδ18O and Dδ13C more or less go in a similar direction between 2.8 and 3.5 Ma at Site 959, with a deepening in the later part (slight offset between oxygen isotopes and carbon and Mg/Ca).*

*Overall I am happy with the changes implemented by the authors and would like to see the manuscript published.*

*However, there are a couple of issues that I would like to authors to reflect on first:*

*Having studied Figure 6, I am not convinced with the following statement (Lines 22-23): 'The tropical thermocline depth evolution of the tropical Atlantic differs from the Pacific, which is characterized by gradual basin-wide shoaling across the Pliocene'. If you compare trends between the Pacific and Atlantic in Figure 6, they seem to more or less the same between ~ 4.2 and 2.8 Ma, with a shoaling between 4.2 and ~3.4 Ma, followed by a deepening until 2.8 Ma. The big contrasting thermocline changes occur earlier between ~4.2 and 5.2 Ma, where that of the Atlantic is deepening and that of the Pacific shoaling. I encourage the authors to make this clear in a further revised manuscript.*

Author reply:

This is a fair point and this needs further clarification in a revised manuscript. The Atlantic-Pacific contrast that we want to point out in this paper is that of the long-term thermocline evolution throughout the Pliocene. This longer timescale is relevant in the discussion of the potential link between tropical thermocline depth and global climate trends. It must indeed be clear to the reader that the Pliocene thermocline evolution in the Atlantic was considerably different than in the Pacific, but that the trends were not consistently opposite (in anti-phase). This was already described in Lines 20-25 (e.g. "periodically divergent Pacific and Atlantic thermocline movements" in Line 25), but we will carefully reconsider passages of the MS that discuss the difference between the Atlantic and Pacific thermocline evolutions and adjust phrasing to make this point clearer.

*Lines 181 to 191. I think that in this part of the discussion the authors should reference the work the work by LeGrande and Schmidt (2006, GRL), where slopes and intercepts for the various regions have been quantified. If using the basin-specific equations create any differences, please discuss this in your further revised manuscript.*

Author reply:

We thank the reviewer for pointing to the paper of LeGrande and Schmidt (2006), which describes the exact same range (0.1-0.3‰/salinity unit) for the tropics as the paper of Conroy et al. (2017) that we cite in line 184, but it also includes data from the tropical Atlantic. We will therefore indeed cite this work here. In the ideal case, this paper would have provided regional slopes for the east and west tropical Atlantic but this is unfortunately not the case. The provided slope for the entire tropical Atlantic is 0.15‰/salinity unit, which falls within the range that we considered in Figure 5 (0.11-0.22‰/salinity unit), additional calculations with this slope will not change the discussion.

*Please provide details about the LOESS smoothing.*

Author reply:

We will specify in the MS that the data were LOESS smoothed in PAST (Hammer et al., 2001) using a span of 0.5.

*Figures in general:*

*Consider making your figures (especially 4 to 7) more compatible for colour-blind individuals.*

Author reply:

We will select more appropriate colours for the vertical bands.

*Figures detailed:*

*Figure 4: why are the axes and labels coloured in a and b? Your colour scheme only fits with c!*

Author reply:

The axes were coloured in an attempt to make it easier to compare the raw records in 4a and b to the smoothed Δx records in 4c, but it seems that this may have had the opposite effect. The axes in 4a and b will therefore be changed to black.

*Figure 5: why are the axes and labels coloured in a and b? The colour scheme only fits with c. Is this figure actually needed? A lot of data is duplicated from Figure 4.*

Author reply:

The axes in 5a and b will also be changed to black, consistent with Figure 4. We prefer to retain both figures to separate results (Figure 4) from discussion (Figure 5) and avoid an overwhelming figure where all records are combined.

*Figure 6: Data from ODP Site 959 and 1000 have considerable gaps. Can you stipple this in the smoothed records to reflect this? Specifically at site 959 between 3.5 and 4.4 Ma there isn't that much data.*

Author reply:

We will stipple the lines as suggested.

---

## Author Comment (AC2)

Dear Editor,

We thank Anonymous Referee #2 for investing additional time to evaluating our revised manuscript. We are appreciative of this positive evaluation and are very happy to discuss the remaining points here and in a revised manuscript.

*The new version of the manuscript by van der Weijst et al. is definitely improved in that additional data have been generated and included, Mg/Ca and stable oxygen and carbon isotopes, into the study. This makes the picture of a deepening of the eastern tropical Atlantic thermocline (TAT) during the warmest part of the Pliocene very convincing. Interestingly, the thermocline shoals again with the intensification of Northern Hemisphere Glaciation. My main point of review on the previous version was that the discussion was not very well developed. And though this has significantly improved I feel this can still be improved. The three possible options that are presented to explain why the TAT shows the same behaviour as in the Caribbean are related to closing of the Panamanian Gateway, temperature changes in the source areas of the thermocline waters and changes in cyclone activity. But the discussion stops with mentioning that neither of these fits very well. I think, however, that the data are convincing enough to make a choice on which explanation the data point to, i.e. related to the closing of Panama and the formation of warm pool-like conditions in the western Atlantic that may have well had their impact as far as the eastern Atlantic. You show in figure 1 the thermocline tilt from the Caribbean to the eastern Atlantic. It would seem very likely to me that a big change like the closing of Panama occurs, that this affects the whole tropical Atlantic.*

Author reply:

We thank the reviewer for their positive evaluation of our revised version and acknowledge the push towards more firm conclusions. We discuss two potential mechanisms for vertical movements of the tropical thermocline: CAS closure, and source water temperature changes, with the latter potentially amplified by a positive feedback with tropical cyclones. To clarify, we do not state that neither of the described scenarios fits well, but rather that the actual mechanisms remain speculative (e.g. lines 24-25 and 281). The reason for this is that data from proxy records and model simulations are currently too limited to confirm or refute the hypothesized mechanisms. However, we prefer the source water/tropical cyclone feedback mechanism over CAS closure, as we express in lines 25-29 and 282-288. That is because climate models simulate a global shallowing of the tropical thermocline in response to CAS closure (Steph et al., 2010; Zhang et al. 2012). The mechanism that drives regional warming in the Caribbean Sea as simulated by Steph et al. (2010), e.g. reduced inflow of cool Pacific surface waters, may not apply to the tropical central and east Atlantic, but this could further be explored in future studies. Likewise, the source water/tropical cyclone hypothesis should be tested in model simulations, and could additionally be supported or refuted by better proxy records from mid-latitudes. As we describe in lines 256-261, available data seems generally consistent with our hypothesis, but it reasonable to believe that currently unknown/undiscussed mechanisms were involved as well. Our mechanistic understanding of oceanic and atmospheric heat transport in relation to thermocline depth is limited, as are our understanding of the geographical and climatological boundary conditions of the Pliocene, including the timing of CAS closure (e.g., Montes et al., 2015; O'Dea et al., 2016).

Because it is clear that much more work is needed to confidently assign a mechanism to the documented tropical thermocline changes in the Atlantic, we would like to refrain from expressing stronger conclusions. Instead, we prefer to present our discussion as a framework for further research. In a revised version, we will more explicitly address what further steps are needed to identify the mechanisms involved in Pliocene thermocline adjustments.

*A second point that I still find not very well developed is the global comparison with other sites. To identify common trends in different basins is a good idea to place the records in a global perspective. But then include some of the compilations that are present, also for the Atlantic like Karas et al. (2017), Bell et al. (2015) or De Schepper et al. (2013, 2014). The location of Site 959 is a great addition to these paper as it indeed shows that it is filling in a blank spot on Pliocene data.*

Author reply:

We thank the reviewer for these suggestions. We carefully reviewed the data presented in these papers and found two southern hemisphere SST records in Karas et al. (2017) that are helpful in the comparison with the thermocline records. These will be included in a revised version of Figure 7. Data from other regions was unfortunately less useful due to limited age ranges and absence of SST data. The additional records further support the discussion in lines 256-261, which will be updated accordingly.

*I'm still a bit confused on why you chose to use Site 1000 for the comparison with Site 959 rather than Site 999. The main reason is that the later part of Site 1000, due to its much shallower waterdepth, is heavily affected by diagenesis (Groeneveld et al., 2006). Especially the Mg/Ca-related temperatures but probably also the isotopes are strongly biased by inorganic precipitates. Site 999 on the other hand comes from a greater waterdepth where diagenesis is not issue, and continuous high-resolution Mg/Ca and d18O records are available. Regarding the number of specimens used for isotopes and Mg/Ca. Were these coming from the same pool of 60 specimens and separated after crushing for either isotopes or Mg/Ca? Or were these separate batches?*

Author reply:

We selected Site 1000, because in contrast to Site 999, there are $\delta^{18}O$ records available of both surface and subsurface-dwelling foraminifera, generated on the same samples. This is needed to calculate the vertical $\delta^{18}O$ gradient ($\Delta\delta^{18}O$) for direct comparison to the complete Pliocene interval at Site 959.

Foraminiferal preservation at Site 1000 was discussed at length by Groeneveld et al. (2008). They conclude that the $\delta^{18}O$ record was not influenced by diagenesis. Instead, amplified $\delta^{18}O$ variability between 4.5-5.6 Ma is interpreted to reflect fluctuating salinity on precessional timescales. They speculate that the Mg/Ca might have been affected by diagenetic overgrowth, but LA-ICP-MS analyses do not support this. Alternatively, they suggest, atypical Mg/Ca values might have been affected by salinity fluctuations. In conclusion, we have no reason to distrust the $\delta^{18}O$ records at Site 1000, especially because we are interested in variability on >20 kyr (precessional) timescales. In a revised manuscript, we will discuss the conclusions of Groeneveld et al. (2008) on the quality of the $\delta^{18}O$ records at Site 1000.

*Lines 132-135: According to Dekens an Atlantic correction is not necessary until 2.8 km waterdepth.*

Author reply:

The data presented in Dekens et al. (2002) was predominantly generated on deeper core tops. The relationship between Mg/Ca and depth above 2.8 km could not directly be determined, and had instead be extrapolated from deeper data. According to Dekens et al. (2002), there are three possible dissolution scenarios in shallower water (see their Figure 8), and they cite evidence from plankton tows that supports dissolution is shallower waters (scenario in Figure 8c). Based on their own data, they can only claim that a dissolution correction is needed in waters >2.8 km. They do, however, not explicitly advice against correction in shallower water. Furthermore, more recent studies do explicitly support a dissolution-correction at shallower tropical Atlantic sites (e.g. Hertzberg and Schmidt, 2013). Regenberg et al. (2014) determined that the critical $\Delta[CO_3^{2-}]$ threshold for Mg/Ca correction is 21.3 $\mu$mol kg$^{-1}$, which is presently located at <2.0 km depth in the east equatorial Atlantic. The communities'

understanding of Mg/Ca calibration and correction is continuously evolving, and the correction factor used in this study could someday be deemed inappropriate. Therefore, we provide raw Mg/Ca values in the supplement so that the records can be recalibrated according to new insights in the future.

*Line 170-172: I agree that propagated errors are getting pretty large, but which alternative do we have? It's the main reason absolute salinities are usually not calculated but we rather stick with relative changes.*

*It would be helpful in the figures to indicate the present-day characteristics, e.g. what is the present salinity difference between surface and thermocline?*

Author reply:

As explained in lines 173-176, we use a semi-quantitative method to calculate relative changes in the vertical salinity gradient. We consider the reconstructed trends reliable, although the magnitude of variation is strongly affected by the assumption of the δw-salinity slope, as illustrated in Figure 5. We cannot indicate the present-day salinity gradient between surface and thermocline in Figure 5, because the y-axis is not calibrated (lines 176-179) and can only be used to infer relative salinity changes through time. We further elaborate on this in lines 181-191, where discuss the possible meaning of the Pliocene Δsalinity record in the context of present-day hydroclimate conditions in the east equatorial Atlantic. For detailed information on the modern situation, the reader is referred to Figure 1 (line 187). In a revised version, we will further elaborate on our choice to calculate relative salinity changes through time.

*Supplement: Put the species names in italics and the isotope numbers in superscript.*

Author reply:

We will do so.

*In conclusion, I think the manuscript still needs more discussion but the addition of new data has improved the story a lot. Along with a clear structure and easy reading I recommend moderate revisions to make this a good contribution to Climate of the Past.*

---

## Author Response (AR1)

**RC1** Referee comments**

Review van der Weijst et al ' Pliocene evolution of the tropical Atlantic thermocline depth. This is a revised manuscript submitted to Clim. Past. Discuss in 2020 which I also reviewed. For this revised manuscript the authors carried out further planktonic foraminifera Mg/Ca measurements and also show the carbon isotopes (new Figures 3 and 4). Thermocline trends inferred from DMg/Ca, D $\delta$ 180 and D $\delta$ 13C more or less go in a similar direction between 2.8 and 3.5 Ma at Site 959, with a deepening in the later part (slight offset between oxygen isotopes and carbon and Mg/Ca).

Overall I am happy with the changes implemented by the authors and would like to see the manuscript published.

However, there are a couple of issues that I would like to authors to reflect on first:

Having studied Figure 6, I am not convinced with the following statement (Lines 22-23): 'The tropical thermocline depth evolution of the tropical Atlantic differs from the Pacific, which is characterized by gradual basin-wide shoaling across the Pliocene'. If you compare trends between the Pacific and Atlantic in Figure 6, they seem to more or less the same between ~ 4.2 and 2.8 Ma, with a shoaling between 4.2 and ~3.4 Ma, followed by a deepening until 2.8 Ma. The big contrasting thermocline changes occur earlier between ~4.2 and 5.2 Ma, where that of the Atlantic is deepening and that of the Pacific shoaling. I encourage the authors to make this clear in a further revised manuscript.

**Author reply:**

We rephrased the header of section 4.2 (line 205 in revised MS) and carefully scanned passages of the MS that discuss the difference between the Atlantic and Pacific thermocline evolution. In combination with the text in the abstract and the figures, it should now be clear that the Pliocene thermocline evolution in the Atlantic was considerably different than in the Pacific, but that the trends were not consistently opposite (in anti-phase).

Lines 181 to 191. I think that in this part of the discussion the authors should reference the work the work by LeGrande and Schmidt (2006, GRL), where slopes and intercepts for the various regions have been quantified. If using the basin-specific equations create any differences, please discuss this in your further revised manuscript.

**Author reply:**

We cite the slope of LeGrande and Schmidt (2006) in line 186. This slope falls within the range that we considered in Figure 5 (0.11-0.22%/salinity unit).

Please provide details about the LOESS smoothing.

**Author reply:**

We specified that the data were LOESS smoothed in PAST (Hammer et al., 2001) in the captions of figures 4-6.

**Figures in general:**

Consider making your figures (especially 4 to 7) more compatible for colour-blind individuals.

**Author reply:**

We selected more neutral colours for the vertical bands.

Figures detailed:

Figure 4: why are the axes and labels coloured in a and b? Your colour scheme only fits with c!

**Author reply:**

**The axes in 4a and b were changed to black.**

Figure 5: why are the axes and labels coloured in a and b? The colour scheme only fits with c. Is this figure actually needed? A lot of data is duplicated from Figure 4.

**Author reply:**

The axes in 5a and b were changed to black, consistent with Figure 4. We prefer to retain both figures to separate results (Figure 4) from discussion (Figure 5) and avoid an overwhelming figure where all records are combined.

*Figure 6: Data from ODP Site 959 and 1000 have considerable gaps. Can you stipple this in the smoothed records to reflect this? Specifically at site 959 between 3.5 and 4.4 Ma there isn't that much data.*

**Author reply:**

We stippled the lines as suggested.

**RC2** Referee comments**

The new version of the manuscript by van der Weijst et al. is definitely improved in that additional data have been generated and included, Mg/Ca and stable oxygen and carbon isotopes, into the study. This makes the picture of a deepening of the eastern tropical Atlantic thermocline (TAT) during the warmest part of the Pliocene very convincing. Interestingly, the thermocline shoals again with the intensification of Northern Hemisphere Glaciation. My main point of review on the previous version was that the discussion was not very well developed. And though this has significantly improved I feel this can still be improved. The three possible options that are presented to explain why the TAT shows the same behaviour as in the Caribbean are related to closing of the Panamanian Gateway, temperature changes in the source areas of the thermocline waters and changes in cyclone activity. But the discussion stops with mentioning that neither of these fits very well. I think, however, that the data are convincing enough to make a choice on which explanation the data point to, i.e. related to the closing of Panama and the formation of warm pool-like conditions in the western Atlantic that may have well had their impact as far as the eastern Atlantic. You show in figure 1 the thermocline tilt from the Caribbean to the eastern Atlantic. It would seem very likely to me that a big change like the closing of Panama occurs, that this affects the whole tropical Atlantic.

**Author reply:**

**Because it is clear that much more work is needed to confidently assign a mechanism to the documented tropical thermocline changes in the Atlantic, we would like to refrain from expressing stronger conclusions. We instead expand on potential approaches for future CAS-related research in lines 250-254.**

A second point that I still find not very well developed is the global comparison with other sites. To identify common trends in different basins is a good idea to place the records in a global perspective. But then include some of the compilations that are present, also for the Atlantic like Karas et al. (2017), Bell et al. (2015) or De Schepper et al. (2013, 2014). The location of Site 959 is a great addition to these paper as it indeed shows that it is filling in a blank spot on Pliocene data.

**Author reply:**

We carefully reviewed the data presented in these papers and found two southern hemisphere SST records in Karas et al. (2017) that are helpful in the comparison with the thermocline records, these were added to Figure 7.

Lines 132-135: According to Dekens an Atlantic correction is not necessary until 2.8 km waterdepth.

**Author reply:**

The communities' understanding of Mg/Ca calibration and correction is continuously evolving, and the correction factor used in this study could someday be deemed inappropriate. Therefore, we provide raw Mg/Ca values in the supplement so that the records can be recalibrated according to new insights in the future.

*Line 170-172: I agree that propagated errors are getting pretty large, but which alternative do we have? It's the main reason absolute salinities are usually not calculated but we rather stick with relative changes.*

It would be helpful in the figures to indicate the present-day characteristics, e.g. what is the present salinity difference between surface and thermocline?

Author reply:

We explain in lines 171-176 why we use a semi-quantitative method to calculate relative changes in the vertical salinity gradient. For detailed information on the modern situation, the reader is referred to Figure 1 (line 187).

Supplement: Put the species names in italics and the isotope numbers in superscript.

Author reply:

Changes were made.

In conclusion, I think the manuscript still needs more discussion but the addition of new data has improved the story a lot. Along with a clear structure and easy reading I recommend moderate revisions to make this a good contribution to Climate of the Past.